# Importance of Prefabrication to Easing Construction Workers’ Experience of Mental Health Stressors

**DOI:** 10.3390/ijerph21091218

**Published:** 2024-09-17

**Authors:** Rasaki Kolawole Fagbenro, Riza Yosia Sunindijo, Chethana Illankoon, Samuel Frimpong

**Affiliations:** School of Built Environment, The University of New South Wales, Sydney, NSW 2052, Australia; r.sunindijo@unsw.edu.au (R.Y.S.); c.illankoon@unsw.edu.au (C.I.); s.frimpong@unsw.edu.au (S.F.)

**Keywords:** construction workers, mental health stressors, poor mental health, prefabricated construction, traditional construction

## Abstract

Construction is widely acknowledged for its socioeconomic contributions, although it is also always considered as a dangerous and incident-prone industry. As a new method of working, prefabrication presents better work environments and other benefits that can potentially improve the safety and mental health of construction workers. This study compares the extent of stressors in traditional and prefabricated construction. Eighty-four construction site and factory-based workers in Australia were surveyed. Prefabricated construction respondents reported less experience of industry-related, management/organisational, and personal stressors. Specifically, the stressors found to be weakened by prefabrication were mental fatigue, work injuries, poor working conditions, unfavourable shift rosters, work overload, and poor work–life balance. Furthermore, the degree of the experience of potential mental health improvement factors such as labour effort efficiency, reduced on-site trade overlap, increased mechanised construction, and less dependence on weather conditions, among others, was significantly higher in prefabrication than in traditional construction. The influence of prefabrication on measures of poor and positive mental health is recommended for further studies, particularly by finding its links with the different groups of construction workers.

## 1. Introduction

The construction industry is one of the largest employers and socioeconomic contributors in developing and developed countries. However, the construction industry is also a significant contributor to the number of work-related physical and mental health challenges. Studies have attributed a poorer state of mental health like anxiety and depression to construction workers than to workers in other industries [1,2]. Poor mental health causes sufferings and adversely affects national productivity through reduced gross domestic products [3,4] and untimely loss of active workers to early retirement, illnesses, and deaths [5]. There is no dearth of studies on identifying and grouping the stressors of poor mental health, and studies have also recommended measures to curb the rising cases of mental health challenges in construction. While there have been records of the success of these interventions [4], the incessant rise of poor mental health in the construction industry is an indication that more needs to be done to achieve a mentally safe work environment for construction workers.

Even though most mental health stressors are rooted in the construction process, its business culture, and physical work environment, commonly recommended mitigating measures, are worker-centric. Some of the measures include discouraging poor mental health-inducing habits such as smoking and drug and alcohol abuse [1], incentivising workers’ adherence to outlined on-site health and safety behaviours and continuous training [6,7]. While the contributions of these measures to achieving improved mental health and safety for workers are acknowledged, they seem to neglect the root causes of mental health challenges that are responsible for most of the unhealthy habits displayed by the workers. Additionally, restricting the fight against poor mental health to worker-oriented programmes only would be supporting the idea that physical and mental pain are unavoidable characteristics of construction [8]. Providing conducive and better physical and psychological work environments for construction workers has been identified as a deep-rooted and lasting approach to achieving improved mental health records for the industry [1,3,4,9].

Prefabricated construction, which is the design, planning, and manufacturing of building components, units, or modules in a controlled manufacturing environment, before being delivered for installation on construction sites, provides better physical and, possibly, favourable psychological, working environments for workers [10]. Unlike the traditional in situ construction, prefabrication ensures the standardisation of construction processes and components [10], and reduces construction time [11], simultaneous construction site preparation, and off-site component fabrication [12] among other benefits. Fagbenro et al. [13] developed a conceptual framework to demonstrate the potential stressor-reducing and mental health-improving potential of the proper implementation of prefabrication. Based on the developed framework, this study aims to empirically confirm the theoretical framework with the achievement of the following objectives:Examine the work-related stressors of mental health among workers, comprising tradespeople and professionals, engaged in traditional and prefabricated methods of construction.Assess the presence and effectiveness of potential stressor-reducing and mental health improvement features in traditional and prefabricated construction methods.

## 2. Literature Review

Although the construction industry, especially in industrialised countries, has recorded significant improvement in the physical safety of its workers, mental health conditions continue to be problematic.

### 2.1. Mental Health Stressors

Several causes or, as they are often called, stressors, have been identified for the prevalent poor mental health record of construction workers. Some of the stressors are related to the individual traits of the workers such as their age [14], gender [15], and cultural and/or religious backgrounds [16] while others originate from management decisions in response to the stressors imposed by the nature of the construction business and process. These stressors were grouped into three based on their sources: industry-related, management/organisational, and personal stressors [13].

#### 2.1.1. Industry-Related Stressors

These are the stressors of poor mental health for construction workers that are inherent in the nature of the construction business, process, and work environment. Fourteen stressors, including work pressure, long working hours, construction business culture, and bodily or musculoskeletal pain, are classified under this category [13].

Pressure has been strongly attributed to construction, particularly because of its competitive bidding nature. Despite the availability of other bidding methods, the lowest bid method remains the most adopted approach to awarding construction contracts [17]. Most construction contracts, because of their low unfavourable award prices are, therefore, laden with cost- and schedule-related performance pressure originating from actions aimed at warding off competition such as inaccurate cost estimates and deliberate underbidding to win a project [18]. While construction companies with adequate resources to conduct business in multiple geographical areas may sometimes deliver projects under the competitive low bids, smaller firms, especially new startups, usually struggle to cope and are constantly under financial and schedule pressure [19]. The contractual pressure on management to deliver quality projects within tight budgets and schedules are eventually transferred to the workers in the form of task overload, low wages, and overtime in a bid to remain in business [20].

Working for longer hours than workers in other industries is a commonly cited issue causing the poor mental health of construction workers [21]. In a bid to deliver projects within schedule, construction programmes are usually contracted to either catch up to lost time from delays or compensate for aggressive and dynamic target schedules imposed by project developers or clients to achieve earlier returns on their investments [22]. In addition to its adverse impacts on project productivity, costs, and quality [22], schedule pressure also poses health and safety concerns for construction workers because of the resulting task overmanning, mandatory overtime, and involuntary shift adjustment to accelerate construction programmes [23]. The business culture and environment in the construction industry, because they are usually characterised by productivity-, cost-, and time-related pressure, therefore, pose adverse physical and mental health conditions for the workers in the form of overtime and poor work–life balance [24].

Although construction operations, especially groundworks and lifting tasks, are reasonably mechanised, there is still the need for human efforts which involves significant physical exertion including bending, twisting, lifting, and prolonged working in awkward postures among other physical demands [25]. Excessive manual handling results in musculoskeletal and body pain, sprains, and other physical injuries for workers [3], all of which have been linked to the poor mental health of construction workers [5].

Other industry-related stressors are psycho-social, physical injuries, fatigue, the unhealthy increase in work speed, work-related physical illness, job insecurity, poor working conditions, nature of construction work, stigmatisation of mental health patients, and job cognitive demand [13]. Psycho-social isolation is common among workers on remote construction projects [26]. Although the advent of digital technologies such as the Building Information Modelling (BIM) has been praised to enhance project performance especially by fostering cost and time-effective workflows [27], Adem et al. [28] identified mental fatigue and psychological pressure as critical side effects of utilising such technologies.

Furthermore, job insecurity, which is a by-product of projects influenced by temporary employment contracts, employment downsizing, economic crises such as inflation, gender and racial discrimination, inadequate education, training, and experience, and technological advancement-powered redundancy, makes construction job availability sporadic or their retention unpredictable [29]. Additionally, the male-dominated work environment [3] promotes stigmatisation against workers with mental health challenges [30] and discourages them from seeking help so they avoid being perceived as emotionally weak by their peers. However, prefabrication, because of its potential to provide a better working environment than in situ construction, could reduce the impact of these stressors [13].

**Hypothesis 1** **(H1):**
*There is a significantly lower experience of industry-related stressors among workers in prefabricated construction than those in traditional construction.*


#### 2.1.2. Management/Organisational Stressors

In reaction to the problems caused by industry-related stressors, construction organisations are compelled to take certain strategic decisions, most of which negatively constitute additional mental health stressors for the workers [13]. There are 12 management/organisational stressors, including interpersonal conflicts, undue and excessive criticism, unfavourable shift rosters, and poor worker support mechanisms.

Construction activities involve personnel of diverse ages, skills, occupations, professions, genders, races, cultures, belief systems, and other distinct features, coming together to achieve project objectives [31]. Such a level of diversity in an industry with fragmentation could promote work-related conflicts stemming from schedule clashes, inadequate co-ordination, ineffective resource management, and poor communication [32]. Poorly managed interpersonal conflicts and persistent scarcity of resources due to cost pressure could deteriorate the mental health of workers [33].

While the process may be repetitive, every construction project is unique, and its success depends much on the clear communication of ever-changing project details within and between the many stakeholders involved [34]. Poor communication and information exchanges add extra stress to the workers because of the additional time and cognitive demand required to complete the tasks at hand [35,36]. Poor information exchange could also lead to construction errors, loss of productive time, wastage of resources, and other challenges [37] that could breed criticism of workers by the management [33].

Due to the usual time- and cost-induced pressure, construction workers are usually working under tight schedules with abnormal work pacing [35]. This forces them to work unfavourable shifts in addition to having an overload of physically and mentally demanding tasks which expose them to risks of mental health challenges [9,14]. Additionally, the industry has seen a rise in the adoption of digital technologies because of their benefits of promoting better project performance through the optimal utilisation of resources and improved procedural efficiencies [27]. However, their excessive usage can adversely affect the workers by eroding their work–life time boundary [24] and expose them to mental fatigue and psychological pressure [28].

Other management stressors are poor workers support [14] and poor feedback mechanisms [38]. The masculine nature of construction sometimes deters mentally distressed workers from seeking help from management. They fear backlash in the form of stigmatisation or being perceived as emotionally weak, which makes most of them deal with the stressors in silence [30]. As a result, the managers in construction organisations are shielded from honest feedback by their employees, and they either provide no mental health support services for the silent workers or provide inadequate support for those who downplay the severity of their mental health challenges. Due to the standardisation benefits of prefabrication, its adoption could appropriately reduce the impacts of management-influenced stressors like ambiguous project instruction and task-related criticisms [13].

**Hypothesis 2** **(H2):**
*There is a significantly lower experience of management/organisational stressors among prefabricated construction workers than among traditional construction workers.*


#### 2.1.3. Personal Stressors

While personal stressors may be viewed as those stressors that are private to an individual worker, the possibility of such stressors being initiated or aggravated by industry-related and management/organisational stressors should not be neglected. Personal stressors are responsible for changes in behavioural patterns or the state of mind of workers, stemming from the nature of construction and management decisions. Reactions to industry-related and management/organisational stressors vary among workers and are dependent on individual traits such as age, gender, marital status, and years of experience, etc. [13]. Personal stressors include age discrimination, workplace harassment, financial difficulties, and poor work–life balance, among others.

Age-based discrimination is suffered by workers of all age groups in construction, whether they are young or old. Young construction workers may be favoured with job retention and promotion because of their physical strength, which is important for tasks involving manual labour [25], while older workers may be preferred because of their experience and social skills [39]. Similarly, female workers suffer gender-based discrimination because of the difficulties they face aligning their lives with the traditional masculine model of the construction industry [40]. Although female construction workers in Australia have better chances of landing office-based roles than site positions, they experience gender-related bullying, discrimination, and harassment more than their male counterparts who are the dominant gender in construction management roles [15].

Wages of construction workers, especially the site workers, are generally considered low. Socioeconomic factors like residency status of workers, alternative or indirect benefits, safety costs, workers’ union activities, and workers’ skillsets including their proficiency with new technologies and equipment, and level of education and/or training, have been cited as causes of low wages [41]. An increase in wages is driven down by inflationary effects, thereby reducing the real value of the wages [42]. These and other factors contribute to financial difficulties and low socioeconomic status, and they have been attributed to prevailing symptoms of poor mental health among construction workers [14,15,38].

Construction workers are prevented from pursuing further learning opportunities because of low awareness of training and further learning programmes that could propel their careers to greater heights. Most of the workers that are aware of these career developmental programmes are discouraged from embarking on them because they seldom get the roles or promotion that are befitting of their updated skillsets [43]. Furthermore, the usual practice of working long hours in construction [40] creates work–life imbalance [21] by reducing the available off-work time to invest in oneself and career development and training [34].

Workers on construction sites are not just of different disciplines and occupations but also of diverse racial, cultural, and religious backgrounds [44]. Interactions between the workers come with some challenges, like language barriers, largely because of the differences in their backgrounds which could lead to racial, cultural, or religiously motivated discrimination of, especially, migrant construction workers [16]. However, prefabrication could alleviate these stressors by virtue of its advantages over traditional construction. Better time performance of prefabrication, for example, could enhance work–life balance and promote the pursuit of further learning and developmental training.

**Hypothesis 3** **(H3):**
*There is a significantly lower experience of personal stressors by prefabricated construction workers than the traditional construction workers.*


### 2.2. Prefabricated Construction

Although prefabrication is not a new concept in construction, its acceptance and application remain underwhelming, especially when its benefits and advantages over the traditional in situ method are widely documented in the literature and corroborated in practice. Known by other names like industrialised building system [45,46], off-site production [47], and off-site manufacturing among others [11], the construction method involves the complete or substantially complete design, fabrication, and pre-assembly of building components and/or units, depending on the degree of prefabrication, in controlled factory settings before they are assembled and transported to sites for final installation and assembly into a single structure with minimal or no extra finishing works [48]. Based on the degree of prefabrication, there are four main types of prefabrication, namely: component manufacture and sub-assembly, non-volumetric pre-assembly, volumetric pre-assembly, and modular construction [10,48].

Building components and/or fixtures that are mass-produced by specialist manufacturers and are available off-the-shelf for construction projects executed with both traditional in situ and prefabrication methods are regarded as component manufacture and sub-assembly [48]. Building components like doors, windows, ironmongery, and electrical switches are some examples and can sometimes be produced on request to suit the special needs of project clients and/or financers. However, it is economically unreasonable to produce them on site [10]. Non-volumetric preassembly is the off-site manufacturing of building components that lack the characteristics of enclosing usable space on their own, like precast columns, beams, and wall panels, etc., but may contain smaller sub-assemblies and are subsequently coupled on-site to form complete building elements [49]. Volumetric preassembly is the design and off-site fabrication and finishing of building units that are capable of enclosing usable space before they are installed within new or existing whole buildings or independent structural frames. Toilet, bathroom, and kitchen pods are common examples of units built with volumetric preassembly, and they are usually prefabricated complete with necessary wares, fittings, and services [50]. Modular construction is the design and off-site or of-the-spot preassembly of building modules or whole building fabric or building units complete with all associated fittings, accessories, finishes, and furniture, etc., before being transported and fixed to the final spot on-site [48,50].

Irrespective of the degree of prefabrication that is implemented, the construction method is explicitly distinguished from the traditional in situ method by its enhanced standardisation of construction processes and preassembly of building components and units under a controlled environment [10,51].

#### Benefits of Prefabrication and Its Stressor-Reducing Potentials

As stated earlier, prefabrication enhances process standardisation, which facilitates better productivity through a more efficient labour and material usage, especially on large and repetitive projects [10]. The repetitiveness of the standardised processes enhances workers’ competencies which could reduce cases of project defects and reworks. Where preassembly is done in a factory environment, prefabrication crew are outrightly protected from the hardship of the weather conditions. Although the on-site erection activities are carried out in the open, the time of exposure to weather conditions by the installation crew is a fraction of the time of weather exposure for traditional construction workers. The segregation of workers into off-site and on-site crews reduces trade overlap [52] while minimal weather exposure and dependence enhances workers’ productivity and promotes the faster delivery of construction projects [45]. Increased productivity can reduce work-related stressors such as rework-induced criticisms, which stem from errors, and promote better interpersonal relationships among the workers [13].

Documentation processes may take longer in prefabrication than in conventional construction, however, the overall project duration is usually reduced because of the simultaneous on-site ground preparations and off-site controlled manufacture of building components and units [12]. Reduction in the on-site time can promote better mental health working conditions through reduced time of weather exposure, flexible shift rosters, reduction in obligatory overtime and workload, better work–life balance, and availability of more personal development time to pursue further training and education [13].

The relationship between physical and mental health has been well established in the literature [3,5]. Prefabrication enhances physical health and safety through easier identification and responding to safety hazards [10], reduction in site congestion-influenced safety risks through reduced trade overlap [53], reduction in the frequency of dangerous tasks, reduction in the number of workers working in dangerous and awkward positions, and better housekeeping practice which drastically limits safety hazards from poor housekeeping and congested sites such as exposed live wires [54]. These and other safety benefits of prefabrication could reduce some stressors of poor mental health [13] such as bodily or musculoskeletal pain [5], physical injuries [3], work-induced physical illness [55], and poor working conditions [34].

**Hypothesis 4** **(H4):**
*There is a significantly higher experience of the potential factors of mental health improvement among prefabricated construction works than among traditional construction methods.*


## 3. Research Methods

Traditional and prefabricated construction methods were compared on the bases of the degree of mental health stressors experienced by their workers and the extent to which the literature established potential stressor-reducing and mental health improvement features are present in both construction methods. Workers from both types of construction in three states/territories in Australia (Australian Capital Territory, New South Wales, and Victoria) were surveyed using a combination of online and paper questionnaire. To be eligible to participate in the survey, participants had to be at least 18 years old, be working on a prefabrication assembly factory or construction site that employed predominantly traditional or prefabricated construction and have a good understanding of the English language. The participant information statement and consent form, which contained brief information about the research objectives, inclusion and exclusion criteria, how the data collected would be used, information on free distress support services, and other information about the research, were provided at the beginning of the survey to prepare the participants for how best to answer the questions if they consented to participate.

Eighty-four (46 traditional, 38 prefabrication) construction professionals and tradespeople, who were selected on a probabilistic stratified sampling method, participated in the survey. The questionnaire had two major sections—the first section contained demographic questions, while the second section addressed the objectives of the research. Knowing that traditional and prefabricated construction do overlap, the methods were defined before the question that sought to categorise the participants by the construction method they identified with. Projects that were executed with mainly in situ construction were classified as traditional, while projects executed with mainly off-site pre-assembly components before they were installed on sites were considered prefabrication. The respondents were then asked to indicate the main construction method that was used on the projects that they participated in. The second section had two sub-sections, with the first containing questions on the experience of mental health stressors by workers from both divides of construction, and the second sub-section explored the mental health improvement features of the construction methods. The variables of the study’s constructs were obtained from a review of the literature on the mental health stressors for construction workers and the benefits and potential mental health improvement features of prefabrication. Section two used a seven-point Likert scale (1 = Never; 2 = Rarely; 3 = Occasionally; 4 = Sometimes; 5 = Often; 6 = Usually; 7 = Always).

Seventeen paper and sixty-seven electronic survey responses were received from a combined 394 electronic and postal mails sent to recruit participants, which represents a 21.32% rate of return.

### Methods of Data Analysis

The reliability of the questionnaire was confirmed with Cronbach alpha’s statistic, with values of 0.70 and above generally regarded as acceptable [56]. Descriptive and inferential statistics were used to analyse the data. The frequency of occurrence was used to summarise respondents’ background information, while the arithmetic mean was used to analyse and rank the research objective variables. The inferential statistics used were the independent samples *t*-test and Pearson’s correlation. The former tests the significance of the difference between the experience of the study’s constructs between the two categories of participants [57] while the latter shows the strength and direction of the relationship between the constructs—mental health stressors and potential factors of mental health improvement [58]. The independent samples *t*-test was used instead of the Mann–Whitney U test because of the normality of the data which was confirmed with skewness and kurtosis. Absolute statistical values less than or equal to 2 and 7 for skewness and kurtosis, respectively, were adopted to represent the normality of the data [56,59]. All statistical analyses were performed with version 28.0.1.0 of the IBM Statistical Package for the Social Sciences (SPSS) Statistics software for Windows, which was released in 2021 by IBM Corporation, Armonk, New York, United States of America.

## 4. Results

The results of the analysed survey data are presented in this section. Background information of the participants is summarised in Section 4.1, while the descriptive and inferential statistical results are presented in the subsequent sub-sections.

### 4.1. Background Information of Participants

Relevant background information of the 84 traditional and prefabricated construction workers are presented in Table 1. Only 17.8% (15) of the participants were female, while the remaining participants were male, thereby confirming the masculinity of the construction industry in Australia and beyond [3,30]. Exactly 75% (63) of the participants were university graduates, with 35 having successfully completed undergraduate degrees, 24 with master’s degrees, and four with doctoral degrees. Others had certificates, including a high school diploma, with vocational certificate holders being represented by 12 participants (14.29%). Over 83% of the participants (70) had construction experience of not more than 15 years. The remaining 14 had at least 16 years of construction experience. Although there are different types of prefabricated construction depending on the metrics used, this study viewed all types of prefabrication as one for easier comparison with the conventional construction method. The two construction methods were fairly represented with 46 and 38 for traditional and prefabrication, respectively.

### 4.2. Reliability and Distribution of Data

The reliability of the data collected was assessed with the internal consistency reliability test of Cronbach’s alpha statistics. The alpha statistics for industry-related stressors, management/organisational stressors, personal stressors, and potential factors of mental health improvement were 0.903, 0.948, 0.932, and 0.945, respectively, indicating internal consistency for the study’s constructs and the reliability of the questionnaire.

Skewness and kurtosis tests were conducted on the study variables to determine the suitability of independent samples *t*-test, which is a parametric test that is applicable to normally distributed data [60], and to confirm the significance of the differences between the means of the participant groups. The statistical values of the skewness and kurtosis, for all the variables of the constructs, fell below the maximum absolute values of two and seven, respectively, which is an indication of the normality of the data distribution.

### 4.3. Workers’ Experience of Mental Health Stressors

The severity of industry-related, management/organisational, and personal stressors was assessed from ‘Never’ for one to ‘Always’ for seven, and the results are presented in Table 2.

#### 4.3.1. Workers’ Experience of Industry-Related Stressors

Work pressure, IR1 (traditional—5.33; prefabrication—4.82; overall—5.10) and long working hours, IR2 (traditional—5.04; prefabrication—4.71; overall—4.89) were the first and second most experienced stressors among both categories of participant. Physical injuries from work incidents (IR5) (traditional—2.83; prefabrication—2.11; overall—2.50) were the least experienced stressor for both prefabricated and traditional construction participants. The mean scores of all the industry-related stressors were higher for traditional construction participants, indicating less exposure to the stressors among prefabricated construction workers. Other critical stressors that had mean scores close to four for prefabricated construction and above four for traditional construction workers were fatigue or tiredness, IR6 (traditional—4.85; prefabrication—4.00; overall—4.46), unhealthy increase in work speed, IR7 (traditional—4.50; prefabrication—3.82; overall—4.19), and job mental (cognitive) demand, IR12 (traditional—4.52; prefabrication—3.89; overall—4.24).

#### 4.3.2. Workers’ Experience of Management/Organisational Stressors

Both prefabricated and traditional construction workers ranked work overload, MS7 (traditional—4.39; prefabrication—3.61; overall—4.04) as the highest experienced management-influenced stressor. However, their perception of the least important stressor differed as prefabricated construction participants ranked unfavourable shift rosters, MS5 (2.21) lowest, while traditional construction participants ranked technology overload, MS6 (2.70) the least stressful situation. Other management-influenced stressors rated high are inadequate provision of job resources, MS2 (traditional—3.91; prefabrication—3.11; overall—3.55) and poor communication of instructions and ideas, MS4 (traditional—3.87; prefabrication—2.92; overall—3.44). The extent of experiencing the management stressors is higher for traditional construction, which is reflected in their higher mean scores than those of prefabricated construction participants for all variables.

#### 4.3.3. Workers’ Experience of Personal Stressors

The most-felt stressor by both traditional and prefabricated construction participants was poor work–life balance, PS7 (traditional—4.43; prefabrication—3.42; overall—3.98). Gender discrimination, PS2 (2.11) ranks the lowest for traditional construction, while religious values conflicts, PS11 (1.89) was the least experienced stressor by prefabricated construction participants. Aside from poor work–life balance (PS7), the experience of other personal stressors among both categories of participants was, at most, on occasional bases, as the mean scores for the variables tended toward two and three. The mean scores of traditional construction participants were, however, higher than the mean scores of prefabricated construction participants, which is an indication of better faring by the latter group.

#### 4.3.4. Mean Variance of Industry-Related Stressors

**Hypothesis 1** **(H1):**
*There is a significantly lower experience of industry-related stressors among prefabricated construction workers than among the traditional construction workers.*


Independent *t*-tests (Table 2) showed significant differences between the means recorded by traditional and prefabricated construction workers for four industry-related stressors. Traditional construction workers experienced more physical injuries from work incidents, IR5 (*p* = 0.035), fatigue or tiredness, IR6 (*p* = 0.018), poor working conditions, IR10 (*p* < 0.001), and stigma or discrimination attached to mental health, IR11 (*p* = 0.028) than their prefabricated construction counterparts. Hence, Hypothesis (H1) was accepted for these four stressors. The results indicate the potential of prefabrication to reduce fatigue or tiredness and promote better working conditions that could reduce stigma towards mental health. Special attention should be paid to both work pressure (IR1) and long working hours (IR2). Both stressors have scores in the region of frequent occurrence in both methods of construction.

#### 4.3.5. Mean Variance of Management/Organisational Stressors

**Hypothesis 2** **(H2):**
*There is a significantly lower experience of management/organisational stressors among prefabricated construction workers than among the traditional construction workers.*


There were significant differences in the extent to which traditional and prefabricated construction workers experienced 10 management/organisation stressors. The stressors were inadequate provision of job resources, MS2 (*p* = 0.026), unclear supervisor’s/management’s directions, MS3 (*p* = 0.008), poor communication of instructions and ideas, MS4 (*p* = 0.005), unfavourable shift rosters, MS5 (*p* = 0,022), work overload, MS7 (*p* = 0.031), undue and excessive criticisms, MS8 (*p* = 0.025), lack of task autonomy, MS9 (*p* = 0.024), lack of participation in decision-making, MS10 (*p* = 0.033), poor workers’ support mechanism, MS11 (*p* = 0.025), and poor feedback mechanism, MS12 (*p* = 0.042). In these cases, Hypothesis (H2) was accepted. It should also be noted that in all significant relationships, traditional construction workers experienced higher stressors than prefabricated construction workers did.

#### 4.3.6. Mean Variance of Personal Stressors

**Hypothesis 3** **(H3):**
*There is a significantly lower experience of personal stressors among prefabricated construction workers than among the traditional construction workers.*


There were significant differences in the mean scores for age discrimination, PS1 (*p* = 0.013), lack of opportunity for future learning, PS6 (*p* = 0.001), and poor work–life balance, PS7 (*p* = 0.013). Again, the mean scores for traditional construction workers were higher than those for prefabricated construction workers, hence Hypothesis (H3) was accepted for the three variables.

### 4.4. Workers’ Experience of Potential Factors of Mental Health Improvement

Twelve distinguishing features and benefits of prefabrication that were reviewed to theoretically improve mental health of workers were presented to the prefabricated and traditional construction participants to rate, on a seven-point Likert scale, the extent to which they experienced them in their work life. The two groups of participants rated improved work health and safety, PB8 (traditional—4.41; prefabrication—5.47; overall—4.89), while less exposure to weather conditions, PB5 (traditional—3.30; prefabrication—4.71; overall—3.94) and labour effort efficiency, PB3 (traditional—3.91; prefabrication—4.61; overall—4.23) were the least rated benefits for traditional and prefabricated construction participants, respectively. Traditional construction participants’ aggregated responses conveyed a moderate enjoyment of construction process standardisation, material usage efficiency, improved construction quality, easier health and safety risk identification, and reduction in on-site dangerous tasks, as the variables (PB1 = 4.30, PB2 = 4.30, PB10 = 4.37, and PB11 = 4.20) had mean scores of approximately four. However, experiences in all the variables of workers in prefabricated construction were better, as the mean scores hovered around five, which translates into ‘often’ in the Likert scale used. Some of the highly ranked variables by prefabricated construction participants were faster completion of construction projects (PB6 = 5.29), easier identification of health and safety risks and/or dangers (PB10 = 5.24), and reduction in frequency of dangerous tasks (PB11 = 5.16). The full results of the analysis are presented in Table 3.

#### Mean Variance of Potential Factors of Mental Health Improvement

**Hypothesis 4** **(H4):**
*There is a significantly higher experience of the potential factors of mental health improvement among prefabricated construction workers than among workers in traditional construction methods.*


As shown in Table 3, there were significant differences in the mean scores of 10 out of the 12 variables (PB3–PB12), with the prefabricated construction participants having a higher magnitude of mean score than the traditional construction participants in all cases. Hence, Hypothesis (H4) was accepted.

### 4.5. Correlation Matrix between the Stressors and the Potential Factors of Mental Health Improvement

A Pearson’s correlation matrix was conducted to examine the strength and significance of the relationships between the latent constructs of mental health stressors and potential factors of mental health improvement. Significant relationships are flagged with asterisks, as shown in Table 4. There were significant positive correlations between industry-related stressors (IRS) and management/organisational stressors (MS), *r*(84) = 0.828, *p* < 0.001, and personal stressors (PS), *r*(84) = 0.721, *p* < 0.001. Similarly, management/organisational stressors (MS) exhibited a significant positive relationship with personal stressors (PS), *r*(84) = 0.733, *p* < 0.001. However, the relationships between the potential factors of mental health improvement (PB) and the stressors (IRS, MS, and PS) were weak, although they may have a potential stressor-reducing capability.

## 5. Discussion of Findings

Although the main classification and comparison criterion of this study is the construction method, analysing the influence of roles and demographic characteristics to understand the impact of prefabrication on mental health is also important. This is necessary because past studies have demonstrated variance in the level of stressors experienced by workers of different roles in the construction industry [34]. Our study could not classify the respondents according to their occupations because of low response rate from the tradespeople. The results presented in this paper are, therefore, strictly based on construction methods and disregard the roles of the respondents in pre-assembly plants or construction sites. This could have affected the significance of the *t*-test scores for some of the stressors. For example, supervisors or site managers, whether they work in traditional or prefabricated construction, may not be in roles that would involve physical exertion as much as tradespeople’s roles do, which could be responsible for the insignificant *p*-value for musculoskeletal pain.

### 5.1. Construction Industry-Related Stressors

The destructive impact of work pressure on the mental health of construction workers of all age categories and genders was highlighted by Hon [33]. Similarly, our research reports the high prevalence of long working hours, which then causes poor work–life balance. The critical experience of fatigue or tiredness, unhealthy increase in speed of work, and high job cognitive demand by traditional construction participants conform to previous studies’ findings [28,34]. Adem et al. [28] attributed mental fatigue of workers to prolonged interaction with machines and industry 4.0 technological tools, while Boschman et al. [34] identified abnormal work speed and work-related mental demand as common industry-related stressors for construction workers, especially those in supervisory roles. Prefabricated construction workers also seem to still experience high work pressure and long working hours, despite the potential of prefabrication to improve process standardisation [10] and time performance [11,12]. However, the statistically significant lower mean score on fatigue, which could be due to the reduction in manual handling and enhanced labour efficiency [10,61], and near significantly low mean scores on job mental demand and unhealthy work speed, confirm the mental health improvement potential of prefabrication.

Despite the relatively low means of IR5 and IR10, indicating that they are not critical stressors, prefabrication still showed better resistance to physical injuries and improved working conditions for the workers than the conventional construction method. Interestingly, it also seems to have the potential to address mental health-related stigmatisation, probably because organisations that adopt this approach are more progressive and treat mental health issues more seriously than their counterparts. In addition, the health and safety benefits of prefabrication, like reduced manual handling [61], reduced on-site construction time [62], less weather exposure [45,63], and reduced trade overlap [52] could have influenced the better performance recorded for the prefabricated construction participants.

### 5.2. Construction Management/Organisational Stressors

Work overload that is frequently experienced by the traditional construction participants aligns with past studies that cite the stressor as a major cause of poor work–life balance and mental health distress [20,33]. Prefabricated construction participants, however, fare significantly better, which attests to the potential of prefabrication to promote good mental health [13]. Although the remaining stressors can be deemed moderate, the better faring of prefabricated construction participants in this regard is important to note and shows that prefabrication can improve the overall working conditions in the construction industry. 

The significantly lower experience of these stressors by the participants in prefabrication confirms the mental health benefits of the construction technique. Reduced work-incited bullying and harassment could be achieved through reduced ambiguity in tasks and role responsibilities [13] and enhanced construction process standardisation [10] which reduces construction mistakes [64] through workers’ increased familiarity with the procedures [10]. Process standardisation, which is achieved with earlier completion of detailed, safer, and more efficient project designs, also reduces the need to continually communicate significant project changes throughout the construction phase, thereby promoting task autonomy, reducing the frequency of task-focused communications on ambiguous tasks, and enhancing communication of standardised information among workers, irrespective of their backgrounds and cadres [13].

### 5.3. Construction Workers’ Personal Stressors

The most critical personal stressor for traditional and prefabricated construction participants is poor work–life balance, although the experience of the traditional group is significantly worse than that of the prefabrication group. The better faring of the prefabricated construction participants corroborates the capability of prefabrication to promote better work–life harmony through a reduction in overall construction time [11], emanating from simultaneous off-site and on-site construction tasks [12], and reduced on-site time and trade overlap [65,66].

Although age discrimination and a lack of further learning opportunities were not rated as critical stressors, prefabricated construction participants had significantly reduced experience of these stressors, which is an indication of the psychological stress-reducing potentials of prefabrication [13]. Design and construction information are finalised and exchanged earlier and in clearer manners to ease standardisation, which is the pivotal feature of prefabrication [10]. The simplicity of the information, repeatability of the standard processes, and reduction in manual handlings enhance competencies and role mastery, reduce construction error-influenced criticisms, and, to an extent, encourage more women to take up construction trades [61]. Similarly, the significantly better exposure to further learning opportunities by prefabricated construction workers is corroborated by the enhanced productivity [10] and reduced workload and work pressure [9,33], achieved through less weather-dependent, faster, and more efficient construction [45,63]. The relatively higher means of the remaining personal stressors for traditional construction affirm the potential of prefabrication to promote better mental health by providing working environments and conditions suitable for psychological and emotional stability [13].

### 5.4. Potential Factors of Mental Health Improvement in Construction Methods

Improved labour output and efficiency affirm the workers’ enhanced familiarity with the standardised processes involved in prefabrication [10], fewer weather interruptions [45], and the substitution of most inconvenient and slower site operations with off-site prefabrication under controlled environments [63]. Additionally, better health and safety records align with the previous literature’s established advantages of prefabrication, like easier and earlier identification of health and safety hazards [10], reduction in the frequency of dangerous site tasks [53], improved housekeeping [54], and reduced incidents from less congested sites [67]. The reduction in the overlap between on-site workers and trade confirms the overall better time performance of prefabrication due to the concurrent execution of off-site pre-assembly and on-site ground works [12]. Building activities that may be unsafe to build in situ become easier and safer to assemble within controlled factory environments [68] with handheld electronic tools and machines, thereby reducing musculoskeletal pain and associated mental distress [61,69]. Finally, the significant higher experience of these factors in prefabrication could potentially reduce the adverse impacts of the stressors and improve the mental health of the workers [13]. While the correlation between the potential factors of mental health improvement (PB) and the stressors of mental were inverse, as expected, the magnitudes of the correlation coefficients were not significant. However, the relationship between the three constructs of the stressors were direct and highly significant.

### 5.5. Limitations of the Study

The major difference in the approach taken in this study on the mental health of construction workers is that it considers the construction methods in which the workers operate. However, the study did not classify the respondents within each construction method according to their trades, occupations, or professions. Although the participants were all either on-site or factory-based personnel on construction projects in Australia, their responses were merged for analysis due to the inadequate number of tradespeople among the participants, without considering that their role differences may have affected the findings. The experience of the stressors and the mental health improvement factors have been confirmed to vary with roles of the workers such as project supervisors and managers providing more contribution in decision making than tradespeople. Further studies should group the workers according to their construction methods and job roles, which may reveal different results.

## 6. Conclusions

The aim of this study was to investigate the potential mental health improvement of prefabrication for construction workers. Unlike existing intervention programmes that are mostly reactive by creating awareness and encouraging distressed personnel to seek help, prefabrication could potentially reduce the impacts of the stressors and provide mental health-friendly work environments and conditions for the workers. Prefabrication was found to be less injury-prone than traditional construction. There were also findings that confirm the significant better faring for prefabrication in terms of work-related tiredness or fatigue, and the promotion of better work environments, which discourages mental health stigmatisation. Prefabrication significantly reduces the severity of management/organisational stressors. This shows that the proper implementation of prefabrication requires some management structure and initiatives that could lessen the management/organisational stressors on the workers. Prefabricated construction workers also experience lower age-related discrimination, have better opportunities for further learning, and better work–life harmony.

As for the potential factors of mental health improvement such as labour effort efficiency, safer and less congested workplaces, better health and safety performance, and less weather-dependent construction, the factors were significantly confirmed as features of prefabrication over the traditional in situ construction method. However, the correlations between the mental health improvement potential factors and the stressors were weak, although the relationships were inverse. This shows the potential of prefabrication to counter the effects of some stressors, which can be strengthened further by combining prefabrication with other proactive interventions for improving the mental health of construction workers.

This research contributes to existing knowledge by investigating the effect of the construction method, in this case traditional and prefabrication, on literature-established mental health stressors and potential protective factors. This is a research topic that has not been empirically investigated before. This research, therefore, is important for demonstrating how prefabrication, which is typically associated with better safety and work environments, can facilitate better mental health, an important issue due to the poor mental health in construction and its socioeconomic implications. Industry stakeholders and policy makers may use the findings of the study to explore the deliberate and wider adoption of prefabrication as a viable means of addressing construction work characteristics that fundamentally cause physical pain and, by extension, mental stress for workers.

## Figures and Tables

**Table 1 ijerph-21-01218-t001:** Demographic profile of survey participants.

S/N	Parameter	Category	Frequency	Percentage
1.	Sex recorded at birth	Female	15	17.85
		Male	69	82.15
2.	Highest level of education and/or training	High school	6	7.14
		Vocational certificate	12	14.29
		Certificate IV	1	1.19
		Diploma	2	2.38
		Undergraduate degree	35	41.67
		Master’s degree	24	28.57
		PhD	4	4.76
3.	Work experience (in years)	1–5	24	28.57
		6–10	27	32.15
		11–15	19	22.62
		16–20	2	2.38
		21–25	4	4.76
		26–30	2	2.38
		31–35	2	2.38
		36–40	2	2.38
		41 and above	2	2.38
4.	Method of construction	Traditional	46	54.76
		Prefabrication	38	45.24

Total number of participants = 84.

**Table 2 ijerph-21-01218-t002:** Mental Health Stressors.

SN	Mental Health Stressors	Traditional			Prefabrication			Overall			*p*-Value
		M	SD	R	M	SD	R	M	SD	R	
**IR**	**Industry-Related Stressors**										
IR1	Work pressure	5.33	1.35	1	4.82	1.16	1	5.10	1.29	1	0.070
IR2	Long working hours	5.04	1.28	2	4.71	1.43	2	4.89	1.35	2	0.264
IR3	Psycho-social isolation (from family and friends)	3.87	1.48	6	3.53	1.37	6	3.71	1.44	6	0.278
IR4	Bodily or musculoskeletal pain	3.65	1.57	7	3.32	1.56	7	3.50	1.56	7	0.329
IR5	Physical injuries from work incidents	2.83	1.76	12	2.11	1.33	12	2.50	1.61	12	**0.035**
IR6	Fatigue or tiredness	4.85	1.61	3	4.00	1.59	3	4.46	1.65	3	**0.018**
IR7	Unhealthy increase in work speed	4.50	1.62	5	3.82	1.57	5	4.19	1.62	5	0.054
IR8	Work-related physical illness	3.04	1.49	11	2.45	1.59	9	2.77	1.61	10	0.091
IR9	Job insecurity	3.09	2.01	10	2.58	1.62	8	2.85	1.85	8	0.212
IR10	Poor working conditions	3.24	1.65	8	2.11	1.31	11	2.73	1.60	11	**<0.001**
IR11	The stigma or discrimination attached to mental health	3.20	1.95	9	2.29	1.71	10	2.79	1.89	9	**0.028**
IR12	Job mental (cognitive) demand	4.52	1.76	4	3.89	1.78	4	4.24	1.79	4	0.110
**MS**	**Management/Organisational Stressors**										
MS1	Interpersonal conflicts with junior and senior colleagues	3.70	1.77	5	3.00	1.64	3	3.38	1.74	4	0.068
MS2	Inadequate provision of job resources	3.91	1.79	2	3.11	1.41	2	3.55	1.67	2	**0.026**
MS3	Unclear supervisor’s/management’s directions	3.54	1.52	6	2.68	1.34	6	3.15	1.49	6	**0.008**
MS4	Poor communication of instructions and ideas	3.87	1.60	3	2.92	1.38	5	3.44	1.57	3	**0.005**
MS5	Unfavourable shift rosters	3.09	1.93	11	2.21	1.40	12	2.69	1.76	11	**0.022**
MS6	Technology overload, e.g., BIM, drones, etc.	2.70	1.63	12	2.24	1.22	11	2.49	1.47	12	0.155
MS7	Work overload	4.39	1.68	1	3.61	1.59	1	4.04	1.68	1	**0.031**
MS8	Undue and excessive criticism	3.43	1.76	8	2.61	1.52	9	3.06	1.70	8	**0.025**
MS9	Lack of task autonomy	3.37	1.77	10	2.58	1.29	10	3.01	1.61	10	**0.024**
MS10	Lack of participation in decision-making	3.37	1.70	9	2.61	1.48	8	3.02	1.64	9	**0.033**
MS11	Poor workers’ support mechanism	3.54	1.88	7	2.66	1.62	7	3.14	1.81	7	**0.025**
MS12	Poor feedback mechanism	3.72	1.72	4	2.95	1.68	4	3.37	1.73	5	**0.042**
**PS**	**Personal Stressors**										
PS1	Age discrimination	2.91	1.77	4	2.00	1.47	7	2.50	1.70	5	**0.013**
PS2	Gender discrimination	2.11	1.43	11	2.08	1.62	8	2.10	1.51	11	0.929
PS3	Workplace harassment	2.26	1.45	10	1.92	1.34	10	2.11	1.41	10	0.273
PS4	Financial difficulties	3.07	1.65	3	2.53	1.45	2	2.82	1.58	2	0.120
PS5	Low socio-economic status	2.59	1.77	6	2.00	1.38	6	2.32	1.62	7	0.099
PS6	Lack of opportunities for further learning	3.26	1.68	2	2.16	1.28	4	2.76	1.60	3	**0.001**
PS7	Poor work–life balance	4.43	1.78	1	3.42	1.87	1	3.98	1.88	1	**0.013**
PS8	Language barriers	2.52	1.66	7	2.13	1.61	5	2.35	1.64	6	0.280
PS9	Racial discrimination	2.41	1.59	9	1.95	1.29	9	2.20	1.47	9	0.150
PS10	Cultural values conflicts	2.85	1,87	5	2.26	1.54	3	2.58	1.74	4	0.127
PS11	Religious values conflicts	2.52	1.94	8	1.89	1.16	11	2.24	1.65	8	0.084

Population = 84. 1 = never; 2 = rarely; 3 = occasionally; 4 = sometimes; 5 = often; 6 = usually; 7 = always. M = mean; SD = standard deviation; R = rank; *p*-values in bold = significant difference.

**Table 3 ijerph-21-01218-t003:** Potential factors of mental health improvement.

SN	Potential Factors of Mental Health Improvement	Traditional			Prefabrication			Overall			*p*-Value
		M	SD	R	M	SD	R	M	SD	R	
PB1	Standardisation of construction processes	4.30	1.64	4	4.61	1.42	11	4.44	1.54	7	0.378
PB2	Efficient materials usage	4.30	1.62	3	4.89	1.56	9	4.57	1.61	5	0.094
PB3	Labour effort efficiency	3.91	1,52	7	4.61	1.57	12	4.23	1.57	11	**0.044**
PB4	Less congestion on site	3.74	1.53	10	4.97	1.79	8	4.30	1.75	9	**0.001**
PB5	Less exposure to weather conditions	3.30	1.40	12	4.71	1.84	10	3.94	1.75	12	**<0.001**
PB6	Faster completion of construction projects	3.91	1.75	8	5.29	1.51	2	4.54	1.77	6	**<0.001**
PB7	Improved quality of construction through enhanced quality control	4.17	1.48	6	5.24	1.55	4	4.65	1.59	3	**0.002**
PB8	Improved work health and safety	4.41	1.65	1	5.47	1.33	1	4.89	1.60	1	**0.002**
PB9	Reduction in on-site trade overlap (better co-ordination among trades or subcontractors).	3.87	1.67	9	5.03	1.55	7	4.39	1.71	8	**0.002**
PB10	Easier identification of health and safety risks and/or dangers	4.37	1.48	2	5.24	1.50	3	4.76	1.54	2	**0.009**
PB11	Reduction in frequency of dangerous tasks	4.20	1.64	5	5.16	1.41	5	4.63	1.60	4	**0.006**
PB12	Reduction in manual works	3.63	1.45	11	5.05	1.72	6	4.27	1.72	10	**<** **0** **.001**

Population = 84. 1 = never; 2 = rarely; 3 = occasionally; 4 = sometimes; 5 = often; 6 = usually; 7 = always. M = mean; SD = standard deviation; R = rank; *p*-values in bold = significant difference.

**Table 4 ijerph-21-01218-t004:** Correlation matrix between the constructs.

(*r*)	IRS	MS	PS	PB
**IRS**	1			
**MS**	0.828 **	1		
**PS**	0.721 **	0.733 **	1	
**PB**	−0.117	−0.046	−0.049	1

Note: ** Correlation is significant at the 0.01 level (2-tailed). **IRS:** Industry-related stressors; **MS:** Management stressors; **PS:** Personal stressors. **PB:** Potential factors of mental health improvement.

## Data Availability

The datasets presented in this article are unavailable for public access due to ethical restrictions from the UNSW’s HREC ethics approval.

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
