# Peer review of "Importance of Prefabrication to Easing Construction Workers’ Experience of Mental Health Stressors"

_ijerph, 2024, doi:10.3390/ijerph21091218_

Round 1
Reviewer 1 Report
Comments and Suggestions for Authors
This is an important area of research. However, this study suffers from methodology issues and lacks theoretical support and implications.
1) Hypotheses are not mentioned until the results section. They should be developed through the literature review.
2) I am doubtful about the validity of the results. As mentioned as one of the limitations of this study, without differentiating the stressors by roles, the comparison findings are invalid as the different roles experience stressors differently.
3) The description of where and why the survey was used is not described in the method. I can see that the survey questionnaire is based on the outcome of the systematic review of the author’s previous publications. For instance, the “potential factors of mental health improvement” measure in this paper are based on the findings of “health and safety benefits of prefabricated construction” of the author’s previous publication, which is more physical safety related than mental health related. Therefore, the validity of the measurement is questionable.
4) The results are more of a descriptive stat and lack theoretical supports and implications.
5) There are two table 3. The second one (line 470), should be table 4 and there is an error on the table caption.
Reviewer 2 Report
Comments and Suggestions for Authors
Paper could benefit from information about the preparedness of respondents to understand the meaning of the stressors. It is not clear what is meant by the terms used to define the stressors and how well were the respondents prepared to answer. Another incomplete information of the study is what are the definitions of the two types of construction studied. Traditional construction has generally prefabricated elements and prefabrication in construction has traditional parts. It is necessary how to decide if a construction is traditional or prefabricated. Paper could benefit from a sample of the survey and of the documents (if any) to explain the stressors that were distributed to respondents.
Reviewer 3 Report
Comments and Suggestions for Authors
The paper is good for publication as such it outlines all needed parameters for a good paper. Grammar is well put together and it will offer reader more interesting approaches in looking at mental stressor.
Can an aspect of implication of the study be added. That is an implication for practice and theory.
